

# Optical and Geometrical Properties of Cirrus Clouds over the Tibetan Plateau Measured by Lidar and Radiosonde Sounding at the Summertime in 2014

Guangyao Dai[1, 2], Songhua Wu[1, 3*], Xiaoquan Song[1, 3], Liping Liu[4]

[1] Ocean Remote Sensing Institute, Ocean University of China, Qingdao, 266100, China

[2] Leibniz Institute for Tropospheric Research, Leipzig, 04318, Germany

[3] Laboratory for Regional Oceanography and Numerical Modeling, Qingdao National Laboratory for Marine Science and Technology, Qingdao, 266237, China

[4] Laboratory of Severe Weather, Chinese Academy of Meteorological Science, Beijing, 100081, China

*Corresponding to*: Songhua Wu (wush@ouc.edu.cn)

**Abstract.** Optical and geometrical characteristics of cirrus clouds over Naqu (31.48°N,92.06°E), the Tibetan Plateau were determined from lidar and radiosonde measurements performed during the third TIbetan Plateau EXperiment of atmospheric sciences (TIPEX III) campaign from July to August 2014. For the analysis of the temperature dependence, the simultaneous observations by lidar and radiosonde were conducted. Cirrus clouds were generally observed ranging from 9.7 to 16.5 km above sea level (a.s.l.), with the cirrus middle temperatures in the range from $-79.7$ to $-26.0\,°C$ . The cloud thickness generally differed from 0.12 to 2.55 km with a mean thickness of $1.22 \pm 0.70$ km and 85.7% of the case studies had thickness smaller than 1.5 km. The retrievals of linear particle depolarization ratio, extinction coefficient and optical depth of cirrus clouds were performed. Moreover, the multiple scattering effect inside of cirrus cloud was corrected. The linear particle depolarization ratio of the cirrus clouds varied from 0.36 to 0.52, with a mean value of $0.44 \pm 0.037$. The optical depth of all the cirrus clouds was between 0.01 and 3 following the scheme of Fernald-Klett method. Sub-visual, thin and opaque cirrus clouds were observed at 9.52%, 57.14% and 33.34% of the measured cases, respectively. The temperature and thickness dependencies on optical properties were studied in detail. A maximum cirrus thickness of around 2 km was found at temperatures between $-60$ and $-50\,°C$ . This study shows that the cirrus mean extinction coefficient of the cirrus clouds increase with the increase of temperature. However, our measurements indicate that the linear particle depolarization ratio has the opposite change tendency along with temperature. The relationships between the presence of cirrus clouds and the temperature anomaly and deep



convective activity are also discussed. The formation of cirrus clouds is also investigated and it has apparent relationship with the dynamic processes of Rossby wave and deep convective activity over the Tibetan Plateau. The cloud radiative forcing is calculated by Fu-Liou model and increases monotonously with the increase of optical depth.

**Introduction**

Cirrus clouds play a significant role in the energy budget and the hydrological cycle of the earth atmosphere system (Webster, 1994). Several studies show that cirrus cover on average 30% of the earth's surface (Nazaryan et al., 2008). The classification of cirrus clouds has been defined on basis of visible observations from the surface. Cirrus clouds are made predominantly or wholly of ice (Lynch, 2002). Only rarely do they contain supercooled liquid cloud droplets at their lower altitudes (Sassen, 2002). Cirrus affects the climate of the whole Earth via two opposite effects; an infrared greenhouse effect and a solar albedo

effect (Giannakaki et al., 2007). The contribution of each effect depends strongly on cirrus optical properties (Zerefos et al., 2003). Thin cirrus clouds usually cause a positive radiative forcing at the top of the atmosphere, while thick cirrus clouds (e.g. opaque cirrus) may produce cooling effect (Stephens and Webster, 1981; Fu and Liou, 1993; Fahey and Schumann, 1999). From the reports of IPCC, significant uncertainties remain regarding the radiative and climate effect of cirrus clouds and cirrus clouds continue to contribute large uncertainty to estimates and interpretations of the Earth's changing energy budget (IPCC,

2013). Since the significant role the cirrus plays, a clear understanding of their optical and geometrical properties is highly essential for climate modeling studies. Moreover, they are critical to understanding feedback processes that regulate or modulate the climate response to forcing.

As one of the best techniques for remotely studying the characteristics and properties of cirrus clouds, lidar will contribute a lot on solving the uncertainties of cirrus. Polarization and Raman lidars have been employed to detect geometrical and optical

properties of cirrus clouds by applying the methods that have been earlier demonstrated by Ansmann et al. (1992). Several key optical parameters such as the extinction coefficient, lidar ratio and depolarization ratio as well as the mid-altitude and the corresponding mid-cloud temperature of cirrus clouds were measured (Sunilkumar and Parameswaran, 2005). The depolarization ratios are considered to be of special importance since are related to microphysics properties of the ice crystals



contained on cirrus clouds, while the optical depth, mid-altitude and mid-temperature play an important role in determining cloud radiative properties.

The optical depth of clouds is a key parameter in radiative transfer computations and therefore considerable effort has been put in its retrieval. The evaluation of optical depth of cirrus clouds was finished through the application of different methods which are already demonstrated in the literature. Most of these methods depend on the solution of the standard lidar equation (Ansmann et al., 1992; Elouragini and Flamant, 1996; Wandinger, 1998). A different technique called 'transmittance method' to determine the optical depth of cirrus clouds from elastic lidar signal has also been developed. It is based on the comparison of the backscattering signals below and above the cloud with assuming that the lidar signals correctly represented the scattering medium (Elouragini and Flamant, 1996; Chen et al., 2002; Giannakaki et al., 2007).

The Tibetan Plateau is a vast elevated plateau in the middle of the Eurasian continent with averaged elevation 4.5 km a.s.l., and it has an important role in the global and regional climate system (Kuwagata et al., 2001). The Tibetan Plateau lies at a critical and sensitive junction of four climatic systems: the Westerlies, the East Asian monsoon, the Siberian cold polar airflow and the Indian monsoon (Wu et al., 2016). Especially given the elevated heat source in summer, the low pressure over the Tibetan Plateau induces a supply of moist, warm air from the Indian Ocean to the continent significantly affecting the general circulation and the Asian Summer monsoon circulation (Wu and Chen, 1985; Wu et al., 2016). Considering the significant thermal effect of the Tibetan Plateau, the statistics of the cirrus clouds over the Tibetan Plateau is crucial. However, the high altitude, lack of oxygen and extreme climate conditions make it difficult to perform long time atmosphere research by ground-based lidar. Even so, for the first time to our knowledge, we conducted the atmospheric measurement by using ground-based high power depolarization and Raman lidar and obtained vertical profiles of atmospheric properties in the Tibetan Plateau during July and August 2014.

In this paper, we briefly describe our lidar system, including the transmitter, receiver and data acquisition. The third TIbetan Plateau EXperiment of atmospheric sciences (TIPEX III) is introduced as well. Furthermore, the methodologies of the determination of the linear particle depolarization ratio, extinction coefficient, optical depth, cloud base/top height and cloud radiative forcing (CRF) are introduced. The correction of multiple scattering inside of cirrus clouds is interpreted as well. One case study of the cirrus clouds optical and geometrical properties in Naqu during the TIPEX III are provided. Combining with

the temperature profiles obtained from concurrent observation by radiosonde, the temperature dependences of such properties are investigated as well. Finally, the longwave fluxes and formation of cirrus clouds are presented and discussed.

## 1. Instrument and experiment

### 1.1 Water vapor, cloud and aerosol lidar (WACAL)

A three-wavelength combined elastic-backscatter Raman lidar, Water vapor, cloud and aerosol lidar (WACAL) is employed to perform continuous observations of suspended aerosol particles and cirrus clouds. The system is based on the second and third harmonic frequency of a compact, pulsed Nd:YAG laser, which emits pulses of 400, 120 and 710 mJ output energy at wavelengths of 355, 532 and 1064 nm, respectively, at a 30 Hz repetition rate. The optical receiver consists of four 308 mm diameter Newtonian telescopes. Five Hamamatsu 10721P-110 photomultipliers and one Hamamatsu G8931-20 APD are used

to detect the lidar signals at wavelengths of 355, 387, 407, 532 (parallel-polarized), 532 (cross-polarized) and 1064 nm. The acquisition system employs a six-channels LICEL transient recorder including analog and photon counting modes. The vertical resolution of the signal is equal to 3.75 m and the temporal resolution is 16 s. Several data pre-processing methods are applied to the raw lidar return signal before retrieving the optical properties. The WACAL was built up successfully in 2013 by the lidar group at Ocean University of China. The details of WACAL has been described by Wu et al. in a separated paper (Wu et

al., 2015).

### 1.2 Radiosonde sounding

The electronic radiosondes of GTS1 type with 76 L-band (Nanjing Bridge Machinery Co., Ltd., China; Qiyun et al., 2012) are used in our research to provide vertical profiles of temperature, pressure and relative humidity. The radiosondes can provide temperature with accuracy of $\pm 0.2\,^{\circ}\mathrm{C}$, relative humidity with accuracy of $\pm 5\%$ and pressure with accuracy of $\pm 1\,\mathrm{hPa}$.

The radiosonde measurement heights can absolutely cover the observation ranges of WACAL.

### 1.3 TIPEX III campaign

The TIPEX III campaign was conducted by a collaboration between Ocean University of China and Chinese Academy of





Meteorology Science. This experiment was carried out during July and August 2014 in Naqu, the Tibetan Plateau and the main

goal was to investigate the water vapor, cloud and precipitation. As an important approach to collect the atmospheric

information, the WACAL system was deployed at Naqu Bureau of Meteorology (31.48 °N, 92.06 °E) from 10 July to 16

August 2014. The measurement site is shown in Fig. 1 with the red dot denoting the lidar site while the yellow star representing

the city of Beijing for the reference. With the help of the WACAL, the optical and geometrical properties of cirrus clouds can

be obtained. To investigate the temperature dependences of the optical and geometrical properties of cirrus, the simultaneous

observations by WACAL and radiosonde are operated. Referring to the results measured by the co-located ceilometer of

VAISALA CL31, the high clouds including cirrus occurred almost at nighttime and before dawn, especially the period beteen

18:00 UTC and 22:00 UTC (Song et al., 2017). Such a temporal distribution of the cirrus clouds is more conducive to the joint

observations by the WACAL and radiosonde sounding. Consequently, only the concurrent measurements by WACAL and

radiosonde at nighttime and before dawn were taken into account.

## 2.   Methodology

In order to get reliable and quantitative results for the optical properties of the cirrus clouds, the methodology is presented and

used in this study.

The SNR of the signal in this channel is calculated from

$$SNR = \frac{P_m(z) - P_{noise} - P_{bg}}{\sqrt{P_m(z)}} \, , \qquad (1)$$

where $P_m(z)$ is the raw signal at the height of $z$ measured by PMTs. $P_{noise}$ and $P_{bg}$ is the noise of the PMTs and the

background of signal, respectively. For the quality assurance and quality control, only the signal of SNR>10 is used for retrieval.

It should be emphasized that in this study, since the altitude of the lidar measurement site is 4508 m a.s.l., the SNR of the

extremely high altitude cirrus clouds signal is still commendable. Hence the retrieval of the cirrus properties benefits a lot.

In this study, only clouds with linear particle depolarization ratios larger than 0.35 (Groß et al., 2011) and with the maximum

optical depth smaller than 3 are considered in order to avoid a possible impact of liquid cloud. With these thresholds, the dust

and volcanic ash layers lofted in high altitude can be filtered out.



Another main aspect is the accurate determination of the cirrus cloud base and top height. The aspect is of major importance for the precise calculation of the optical and geometrical properties of cirrus clouds.

## 2.1 Calibration of the linear particle depolarization ratio

WACAL system provide the linear particle depolarization ratio measured at 532 nm. For the precise measurement of the linear particle depolarization ratio, an effective and accurate calibration method called "±45°-calibration" is used in this paper (Freudenthaler et al., 2009). In this method, the influences of the gain ratio, offset angle of the receiver module and the properties of the polarizing beam splitter (PBS) are calibrated by using half wave plate. After the calibration, the linear particle depolarization ratio can be represented as a function of component variables including measured ratio $m(r)$, gain ratio $G$, offset angle $\varphi$, the reflectivities and the transmittances ($T_P$, $T_S$, $R_P$ and $R_S$) of PBS by

$$\delta^v(r) = \frac{m(r)T_P - GR_P + (m(r)T_S - GR_S)\tan^2\varphi}{GR_S - m(r)T_S + (GR_P - m(r)T_P)\tan^2\varphi} \quad \text{and} \tag{2}$$

$$\delta^p(r) = \frac{(1+\delta^m)\delta^v(r)R - (1+\delta^v(r))\delta^m}{(1+\delta^m)R - (1+\delta^v(r))}, \tag{3}$$

with $\delta^v$ is the volume depolarization ratio, $\delta^m$ is the depolarization ratio for molecule which is set as 0.0036 for 532 nm and $R$ is the backscatter ratio.

## 2.2 Determination of the extinction coefficient

Based on the elastic-backscatter signal at wavelength of 532 nm, the particle extinction coefficient profile can be calculated by using the method of Fernald (Fernald, 1984; Ansmann et al., 1992) and can be written as

$$\alpha_{\lambda_0}^{aer}(z) + \frac{S_{\lambda_0}^{aer}(z)}{S_{\lambda_0}^{mol}}\alpha_{\lambda_0}^{mol}(z) = \frac{S_{\lambda_0}^{aer}(z)P_{\lambda_0}(z)z^2 \exp\{-2\int_{z_0}^{z}[\frac{S_{\lambda_0}^{aer}(\zeta)}{S_{\lambda_0}^{mol}}-1]\alpha_{\lambda_0}^{mol}(\zeta)d\zeta\}}{\dfrac{S_{\lambda_0}^{aer}(z_0)P_{\lambda_0}(z_0)z_0^2}{\alpha_{\lambda_0}^{aer}(z_0)+\dfrac{S_{\lambda_0}^{aer}(z_0)}{S_{\lambda_0}^{mol}}\alpha_{\lambda_0}^{mol}(z_0)} - 2\int_{z_0}^{z}S_{\lambda_0}^{aer}(\zeta)P_{\lambda_0}(\zeta)\zeta^2\exp\{-2\int_{z_0}^{\zeta}[\frac{S_{\lambda_0}^{aer}(\xi)}{S_{\lambda_0}^{mol}}-1]\alpha_{\lambda_0}^{mol}(\xi)d\xi\}d\zeta}, \tag{4}$$

where $S_{\lambda_0}^{mol} = 8\pi/3$ sr and $S_{\lambda_0}^{aer}(z)$ are the extinction to backscatter ratios for Rayleigh and particle scattering, respectively. In this paper, $S_{\lambda_0}^{aer}(z)$ is assumed as a constant and independent of the range. For the cirrus clouds discussed in this study, based on the pioneered study operated by He et al., the $S_{\lambda_0}^{aer}(z)$ is chosen as 28 sr.



The method for determination of the $z_0$ is known as "Minimum value method", which is also described in Wu et al. (2015). In this method, the $z_0$ is corresponding to the height $z$ where the value of $P_{\lambda_0}(z)z^2 \big/ \beta_{\lambda_0}^{mol}(z)$ is minimum. As for the determination of the $\alpha_{\lambda_0}^{aer}(z_0)$, as the reference height $z_0$ is fixed, the iteration of Fernald algorithm is taken into consideration. The $\alpha_{\lambda_0}^{aer}(z_0)$ starts with the value of $4\times10^{-9}$ m$^{-1}$. From the derived extinction coefficient result, the mean value $\alpha_{\lambda_0,mea}^{aer}(z_0)$ is computed. For each iteration step, the input value of $\alpha_{\lambda_0}^{aer}(z_0)$ is increased by 10% until the relative difference between $\alpha_{\lambda_0}^{aer}(z_0)$ and $\alpha_{\lambda_0,mea}^{aer}(z_0)$ is less than 5% (Althausen et al., 2009).

## 2.3 Multiple scattering correction

The multiple scattering (MS) returns of elastic signal from cirrus clouds should be considered. The amount of the MS returns relies on the forward diffraction peak in the phase function (Comstock and Sassen, 2001). The MS factor is the function of laser penetration, cloud range, FOV and mean effective radius for ice particles (Platt et al., 1987; Sassen and Cho, 1992; Trouillet et al., 1999; Chen et al., 2002). However, there is no obvious dependence of the forward diffraction peak on ice crystal shape, which was reported by Macke et al. (1996). In this study, for accurately correcting the MS influence, we perform a full treatment of multiple scattering following the model of Hogan (2008). The model is also introduced and applied by Seifert et al. (2007), Kienast-Sjögren et al. (2016) and Gouveia et al. (2017). In this model, the typically operated FOV of 1.3 mrad and the full divergence angle of 0.5 mrad in WACAL is input. Moreover, the particle size of cirrus clouds which is modelled according to the method in Krämer et al (2016) is applied into the multiple scattering model to correct the effect in cirrus clouds as well.

## 2.4 Extraction of the cloud base and top heights

In order to determine the cloud base and top heights, an algorithm that combines "differential zero-crossing algorithm" (Pal et. al., 1992) and "threshold algorithm" (Winker and Vaughan, 1994) is used when the clouds can be penetrated by laser. In most of cases, cloud heights can be defined directly from these zero crossings of the first derivative of backscatter intensity $dP(z)/dz$. However, to exclude interference of spurious zero crossings, the $dP(z)/dz$ is calculated via least-squares fit by using a multipoint window that slides through successive points from $z_0$ to the end of the lidar profile. Additionally, based





on the noise in the recorded lidar signal, a series of range-dependent thresholds are defined. Only the signal excursion above the corresponding threshold value is identified as a cloud. By this algorithm, cloud layers apparent in the return signal are identified. However, if the lower clouds were too thick to be penetrated, the higher clouds will not be detected and cannot be identified (Wu et al., 2015, Song et al., 2017). During this campaign, the cloud base heights determined by the WACAL and

that from the co-located VAISALA CL31 ceilometer measurements were compared and the results were reported by Song et al. (2017). It shows a good consistency with each other. In this study, in order to ensure that the lidar is able to detect the cloud top, the lidar signal above the cirrus is checked. When the signal still has a tendency to be attenuated, the cirrus top is ensured. Moreover, since the lidar is restricted when the optical depth is bigger than 3, only cirrus clouds with the whole path optical depth smaller than 3 are taken into consideration.

**2.5 Cloud radiative forcing**

Cloud radiative forcing is a parameter that has been used extensively to study cloud-radiation interactions. The Fu-Liou radiative transfer model is used here to estimate CRF. The model is available freely online through some subsequent modifications (https://www-cave.larc.nasa.gov/cgi-bin/fuliou/runfl.cgi). It is a delta-four steam radiative transfer scheme with 15 spectral bands from 0.175 to $4.0\,\mu m$ in shortwave (SW) and 12 longwave (LW) spectral bands between 2850 and $0\,cm^{-1}$.

The corrected k-distribution method is used to parameterize the non-gray gaseous absorption by $H_2O$, $CO_2$, $O_3$, $N_2O$, and $CH_4$ (Fu and Liou, 1993). The Fu-Liou model inputs here are visible optical depth, phase, cloud top and base pressure, and particle size. The optical depth can be obtained from above cirrus optical properties retrieval. Cloud phase is an input as either WATER or ICE phase, which affects the input of cloud particle size. Here the cirrus clouds are chosen as ICE phase. The cloud top and base pressures can be obtained from radiosonde data.

**3. Results and discussion**

During the TIPEX III, 21 measurement cases of cirrus clouds were performed, 17 of them were observed at night-time while the rest were detected before dawn. In this section, the optical and geometrical properties of these cirrus clouds are presented. With the temperatures measured by radiosondes that are launched at Naqu meteorological bureau twice per day, at 0000 UTC





and 1200 UTC (LST – 8 h), the temperature dependence of optical and geometrical properties is investigated and discussed. Furthermore, the relationship of the appearance of cirrus clouds and the temperature anomaly, as well as the cirrus radiative forcing are investigated.

## 3.1 Case study

For detailed study of the optical and geometrical properties of cirrus clouds, we present one measurement case on 4 August 2014 in Fig. 2. During this measurement period, there are two cirrus clouds layers were observed.

According to Fig. 2(a) and (c), the developments of linear particle depolarization ratio and extinction coefficient inside of the cirrus clouds are expressed in detail with a temporal resolution of 80 s. For instant, one depolarization ratio profile and one extinction coefficient measured at 21:00 LST are presented in Fig. 2(b) and (d). It should be emphasized that only the
depolarization ratios inside of cirrus clouds are plotted and the extinction coefficients are shown with and without multiple scattering correction. Due to the inherent inhomogeneity inside of the cirrus clouds, the linear particle depolarization ratio and extinction coefficient vary as the cirrus evolves and ice crystal habit changes. The value of linear particle depolarization ratio under 7 km above ground level (AGL) is generally smaller than that of cirrus higher. Conversely, the value of extinction coefficient under 7 km AGL is larger than that of cirrus higher. The results may be explained by the temperature dependence
changes of the crystal shape, shape ratios, number concentration and size distribution of ice crystals in cirrus clouds. With the increase of the height, the temperatures gradually decrease. According to the previous work (Chen et al., 2002, Sunilkumar and Parameswaran, 2005, Seifert et al., 2007, He et al., 2013), temperature has great impact on linear particle depolarization ratio and extinction coefficient by dominating the crystal shape, number concentration and other microphysics properties of cirrus. Furthermore, the cloud base and top height of the first cirrus layer are $6.02 \pm 0.60$ km and $8.56 \pm 0.69$ km, respectively.
The corresponding linear particle depolarization ratio and mean extinction coefficient are $0.39 \pm 0.02$ and $0.171 \pm 0.084$ km$^{-1}$, respectively. The mean temperature of this layer is $-39.8 \pm 4.0\,°C$. The cloud base and top height of the second cirrus layer are $9.70 \pm 0.39$ km and $11.77 \pm 0.76$ km, respectively. The corresponding linear particle depolarization ratio and mean extinction coefficient are $0.40 \pm 0.02$ and $0.07 \pm 0.04$ km$^{-1}$, respectively. And the mean temperature of this layer is $-67.0 \pm 4.4\,°C$.

## 3.2 Geometrical characteristics and temperature dependence



Cloud thickness and structure are important factors in determining the cirrus cloud properties. Based on the measurements of WACAL, the structures of cirrus clouds were provided. In Fig. 3, we present the cirrus structure and the vertical distribution of the occurrence frequency of cloud base/top heights measured during the TIPEX III. The abscissa axis coordinates in style of "MMDD-No" indicate the measurement date and the serial number of cirrus on that day. Fig. 3(a) shows the cirrus clouds

structure measured by WACAL over Naqu and makes the details of cirrus clouds much clearer. The fluctuation of cloud base/top heights were also presented. Fig. 3(b) presents the occurrence frequency of cloud base and top heights at 1 km altitude interval. The maximum frequencies of the cirrus cloud base and top height were 33.34% at height of 6-7 km AGL and 33.34% at height of 7-8 km AGL, respectively. The cirrus base height ranged from 5 to 12 km AGL, with a mean value of $7.53 \pm 1.79$ km AGL. Meanwhile, the cirrus top height ranged from 5 to 13 km AGL, with a mean value of $8.75 \pm 1.83$ km AGL.

Histograms of cirrus cloud middle height (geometric center height) and cirrus cloud mean temperature are presented in Fig. 4. The middle heights of the cirrus clouds are generally observed in the Tibetan Plateau region from 5 to 12 km AGL and 66.6% of them are located between 6 and 8 km AGL. According to Fig. 4(b), the maximum frequency of the cirrus middle temperatures is 33.34% between -50 ℃ and -40 ℃ and the mid-cloud temperatures range from -79.7 to -26.0 ℃ with a mean value of $-48.0 \pm 15.0$ ℃.

From Fig. 5, it can be found that the cirrus thickness distribution has a skewed normal distribution. 85.7% of the case studies have thickness smaller than 1.5 km and the cloud thickness ranges from 0.12 to 2.55 km with a mean thickness of $1.22 \pm 0.70$ km.

## 3.3 Optical properties and temperature dependence

In our study, the extinction coefficient, the linear particle depolarization ratio and optical depth are calculated. We applied the

methods which were briefly presented in the methodology section to all the case studies of cirrus clouds during the TIPEX III over Naqu. In this study, the linear particle depolarization ratio and cirrus mean extinction coefficient and the corresponding statistical uncertainties are calculated. The depolarization ratio ranges from 0.36 to 0.52, with the cirrus mean extinction coefficient differs from $5.40 \times 10^{-5}$ m$^{-1}$ to $4.00 \times 10^{-4}$ m$^{-1}$.

For further investigation of the extinction coefficient and optical depth, we present the distribution and histogram of the



cirrus clouds optical depth in Fig. 6. Several investigations on the statistics of cirrus clouds have been accomplished. The threshold of 0.03 for optical depth measured at the wavelength of 694 nm was first proposed by Sassen and Cho (1992) in order to separate cirrus clouds to visible and sub-visual. Goldfarb et al. (2001) found that ~20% of total cirrus cloud occurrences are sub-visual for a mid-latitude lidar station in France. Reichardt (1999) has found that 70% of cirrus clouds were classified

as sub-visual or optically thin ice clouds (optical depths bellow 0.3). Sassen and Campbell (2001) have presented a mid-latitude climatology showing that optical depth of cirrus clouds is greater than 0.3 for ~50% of detected cases.

  In this study, according to Fig. 6, the optical depth of all the cirrus clouds is between 0.01 and 3. We have found that 9.52% of cirrus clouds are sub-visual (optical depth<0.03), 57.14% of cirrus clouds are optical thin (0.03<optical depth<0.3) and 33.34% of cirrus clouds are opaque (optical depth>0.3). In our study, 66.66% of the cirrus clouds during this Tibetan Plateau

campaign are sub-visual or optically thin, which is distinctly higher than the occurrence frequency of 50% that measured by He et al (He et al., 2013).

  For the purpose of investigating the temperature dependence of linear particle depolarization ratio and mean extinction coefficient distribution, we drew the scatter diagrams of linear particle depolarization ratio, mean extinction coefficient and temperature, cirrus middle height.

In Fig. 7, the linear particle depolarization ratios dependences on temperature and cirrus middle height are presented. The bars present the statistical uncertainties of linear particle depolarization ratio while the black lines are the fitting curves. From Fig. 7, the linear particle depolarization ratio of cirrus varies from 0.36 to 0.52. With the increase of the temperature, the values of the linear particle depolarization ratio is gradually reduced, which is in agreement with previous papers (Chen et al., 2002, Sunilkumar and Parameswaran, 2005). Depolarization ratio is sensitive to crystal shape and shape ratios (Noel et al., 2002).

With the extremely low temperature, the formation of crystals with large mean sizes and more complex form with irregular structures may initiate (He et al., 2013). We propose that with the decrease of the temperature and the increase of cirrus middle height, the irregular type ices with larger size and greater distortion are formed, which result in the increase of linear particle depolarization ratio of the cirrus clouds. According to Fig. 7, it can be seen that the linear particle depolarization ratio with higher temperature and lower altitude is much more sensitive to the ambient temperature. Under the condition of low

temperature, since the existence of the stability of the larger cirrus particle radius, the temperature sensitivity is found gradually





weak with the decreasing temperature.

Figure 8 shows the mean extinction coefficient dependences on temperature and cirrus middle height. The bars present the

statistical uncertainties of extinction coefficient while the black lines are the fitting curves. From Fig. 8, as expected, the mean

extinction coefficient decreases with decreasing temperature for the measured cirrus clouds over the Tibetan Plateau, which is

in agreement with previous papers (Seifert et al., 2007, He et al., 2013). For a certain wavelength of 532nm in WACAL, the

extinction coefficient is essentially influenced by the number concentration and size distribution of ice crystals in cirrus clouds.

Although the cirrus radiative properties depend on cloud microphysics, it is the ambient temperature that governs the radiative

transfer. It dominates the microphysics such as number concentration and size distribution. Platt et al. have shown the cirrus

extinction coefficient is a well monotonic function of atmospheric temperature and yields the follow equation of

$\alpha = aT^2 + bT + c$  (Platt, 1988). Wang and Sassen developed a second-order polynomial function for describing the

temperature dependence of cirrus extinction coefficient by using the lidar and radar data (Wang and Sassen, 2002). Here we

applied the second-order functional forms in representing the temperature dependence of cirrus mean extinction coefficient

and the formula is  $Ext.cf. = 4.49 \times 10^{-8} T^2 - 8.14 \times 10^{-6} T + 4.40 \times 10^{-4} \ (m^{-1})$ . Moreover, we also calculated the relationship

between the cirrus mean extinction coefficient and the height AGL and the fitting curve is presented in Fig. 8(b). Due to the

stability of the larger cirrus particle radius in the extremely low temperature, the temperature sensitivity of cirrus mean

extinction coefficient is gradually weakened with the decreasing temperature. Consequently, temperature has a more prominent

impact on extinction coefficient for warmer and lower cirrus.

**3.4  Cirrus occurrence and temperature anomaly**

The temperature anomalies during July and August 2014 over Naqu is provided in Fig. 9. The simultaneous tropopause heights

and cirrus base/top heights were provided as well. From this figure, most of the cirrus were below tropopause except the cases

on 10 and 18 July 2014. But we still took them into consideration since the cirrus top heights of these two cases were smaller

than 1.5 km above the tropopause. The warm color corresponds to the positive deviation while the cool color corresponds to

negative deviation. The cirrus cloud base and top heights are also plotted in Fig. 9. The pink star symbol denotes the cirrus

base height and the black square symbol represents the cirrus top height. The lidar observation can be separated in six stages:



10 to 11 July, 18 to 22 July and 31 July to 08 August for the cirrus appearance stages and 12 to 17 July, 23 to 30 July and 09 to 16 August for the cirrus blank stages. During the 3 cirrus appearance stages, the temperatures were higher than the average value, which indicates that the formation of the cirrus clouds may be related to the local ground heating and deep convective activity over the Tibetan Plateau. In this period, cold perturbations were found at height of about 12 km AGL. During the 3 cirrus blank stages, oppositely, the temperatures were lower than the average. Consequently, it can be found that the tropospheric temperatures with the presence of cirrus were about 5 °C higher than that with no cirrus clouds. Based on the appearance of the colder and warmer air masses at the height of above 12 km AGL (16.5 km a.s.l.), the Rossby waves can be recognized at the same height in the tropospheric layer and with a duration cycle of about 10 days. Considering the inherent characteristics of the strong local heating and the atmospheric instability in the Tibetan Plateau, it is proposed that the appearance of cold perturbation in the tropopause layer or beyond the troposphere may enhanced the convective activities above the ground. Hence it is more conducive to the formation of the cirrus clouds.

For researching the significant impact of the radiative forcing and climate effect caused by cirrus clouds, we calculated the longwave fluxes of cirrus clouds based on the Fu-Liou Model. Since the microphysical properties such as mean effective radius for ice particles is unavailable from WACAL system, we modelled the ice particle size by the method described in Krämer et al. (2016). Fig. 10(a) shows that the cirrus radiative forcing $F_c$ increases monotonously with the increase of optical depth. $F_c$ is smaller than $6 \, \mathrm{Wm^{-2}}$ for sub-visual cirrus while for thin cirrus, the values of $F_c$ lies between $6 \, \mathrm{Wm^{-2}}$ and $25 \, \mathrm{Wm^{-2}}$. As for the opaque cirrus, the value of $F_c$ is greater than $25 \, \mathrm{Wm^{-2}}$. Comstock et al. (Comstock et al., 2002) estimated that a cloud becomes significant in term of outgoing longwave radiation (OLR) when the cloud forcing is approximately $10 \, \mathrm{Wm^{-2}}$. According to Fig. 10(a), the cirrus clouds in this study with OD>0.095 have a significant cloud forcing and the value of OD is slight bigger than that reported by Comstock et al. Moreover, the relationships between cloud thickness, mean extinction coefficient and radiative forcing are shown in Fig. 10(b) and (c). It is found there is no obvious direct dependency among these parameters.

## 4.   Summary and conclusion



This study provides an analysis of the geometrical and temperature characteristics as well as the optical properties and cloud radiative forcing of cirrus clouds over the Tibetan Plateau. Our results show that cirrus clouds are occurring between 9.7 to 16.5 km a.s.l., with temperatures ranging from -79.7 to -26.0 ℃. And the mean cirrus middle temperature is -48.0 ± 15.0 ℃. The cloud thickness ranges from 0.12 to 2.55 km with a mean thickness of 1.22 ± 0.70 km. The linear particle depolarization

ratio differs from 0.36 to 0.52, with a mean value of 0.44 ± 0.037.

According to the measured optical depth, the cirrus clouds are classified as sub-visual cirrus (9.52%), optical thin cirrus (57.14%) and opaque cirrus (33.34%). From the results, the fitting curves about linear particle depolarization ratio, mean extinction coefficient depending on temperature and cirrus middle height are drawn. The corresponding gradients are also calculated and are presented in Fig. 7 and Fig. 8.

According to the temperature anomaly during the July and August in 2014, the formation of cirrus clouds has apparent relationship with the dynamic processes of Rossby wave and deep convective activity over the Tibetan Plateau.

The cloud radiative forcing of sub-visual, thin and opaque cirrus was calculated respectively and they increase gradually along with the optical depth.

The dependence of cirrus mean optical and geometrical properties on temperature with an interval of 10 ℃ is also provided

in Fig. 11. According to Fig. 11 (a) and (d), with the decrease of the temperatures, the developments of linear particle depolarization ratio and mean extinction coefficient show opposite change tendencies. With the changes of the temperature-dependent number concentration and size distribution of ice crystals in cirrus clouds, an indication that the mean extinction coefficient increases with increasing temperature is found. In contrast, the linear particle depolarization ratio which is sensitive to ice crystal shape and shape ratios increases along with the decrease of temperature. It can also be noted from Fig. 11(b) and

(c) that the mean cirrus thickness and optical depth have the similar tendencies. Accordingly, the maximum optical depth and cloud thickness are both found at the temperatures between −60 and −50 ℃.

*Acknowledgements.* This work was supported by the National Natural Science Foundation of China (NSFC) under grant 41375016 and 91337103, by the China Special Fund for Research in the Public Interest under grant GYHY201406001. The authors are very grateful to Patric Seifert from Leibniz Institute for Tropospheric Research, Germany for providing us



comments about cirrus filters, quality control and multiple scattering. We also thank Diego A. Gouveia from University of São Paulo, Brazil for discussion and providing scripts for correcting multiple scattering of cirrus. We appreciate the contribution made by Meteorological Bureau of Tibet Autonomous Region, Naqu Bureau of Meteorology. We also thank Dongxiang Wang, Xiaochun Zhai and Qichao Wang for their help with the observations and discussions.

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



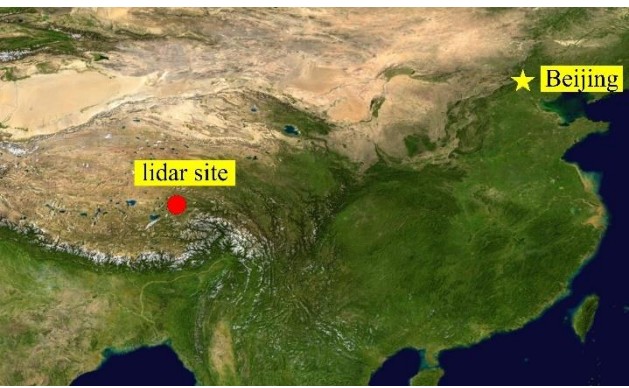

**Figure 1: The measurement site of the TIPEX III. The red dot denotes the lidar site while the yellow star represents the Beijing city.**

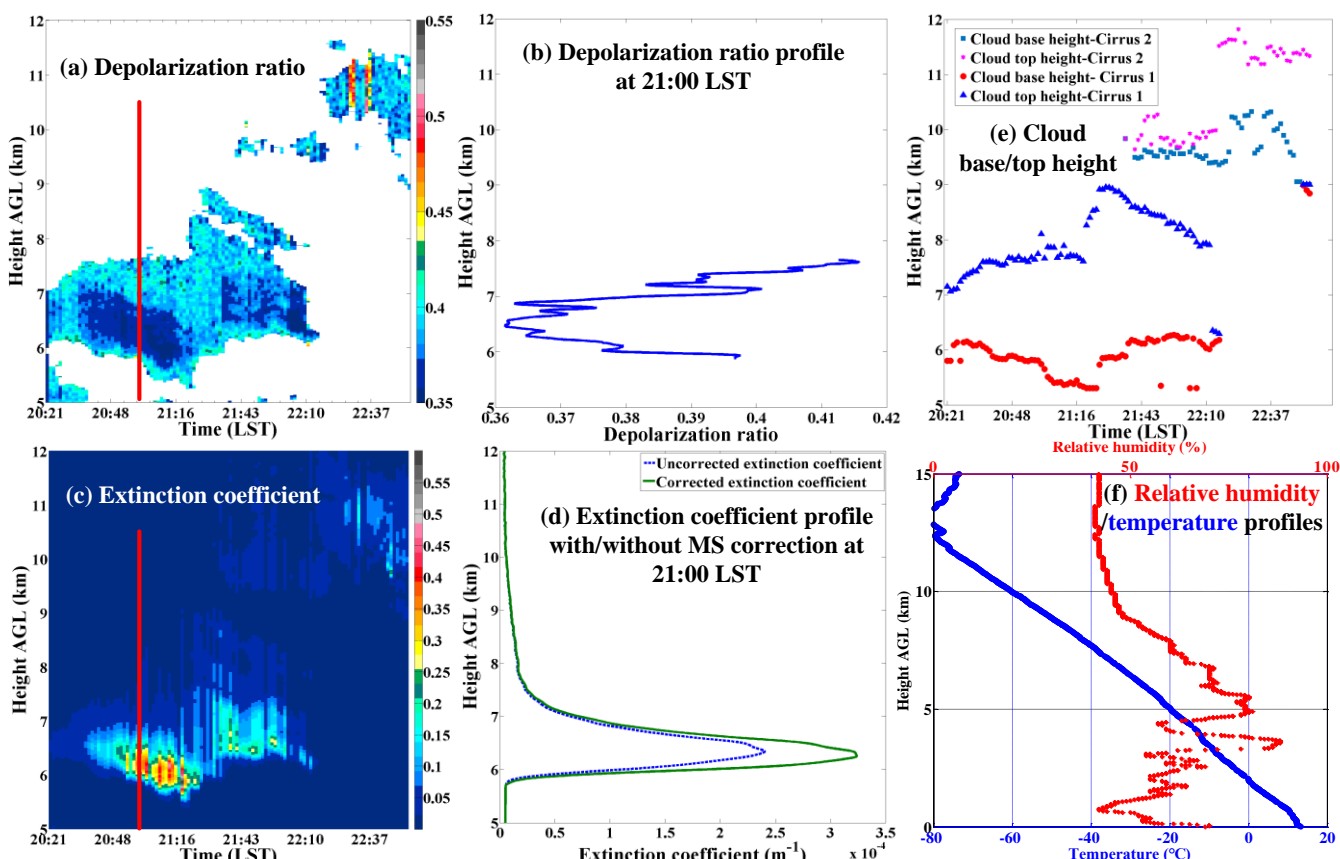

**Figure 2: Time serials of (a) depolarization ratio, (c) extinction coefficient, (e) cloud base/top heights on 04 August 2014. In this**

5 **figure, (b) one depolarization ratio profile and (d) one extinction coefficient profile measured at 21:00 LST are provided, as well as**

**the (f) relative humidity/temperature. Please note that only the depolarization ratios inside of the cirrus are provided. In the Fig. (d),**

**the extinction coefficient with multiple scattering correction and without the correction are plotted and the results are denoted by**

**dark green solid line and blue dashed line, respectively.**





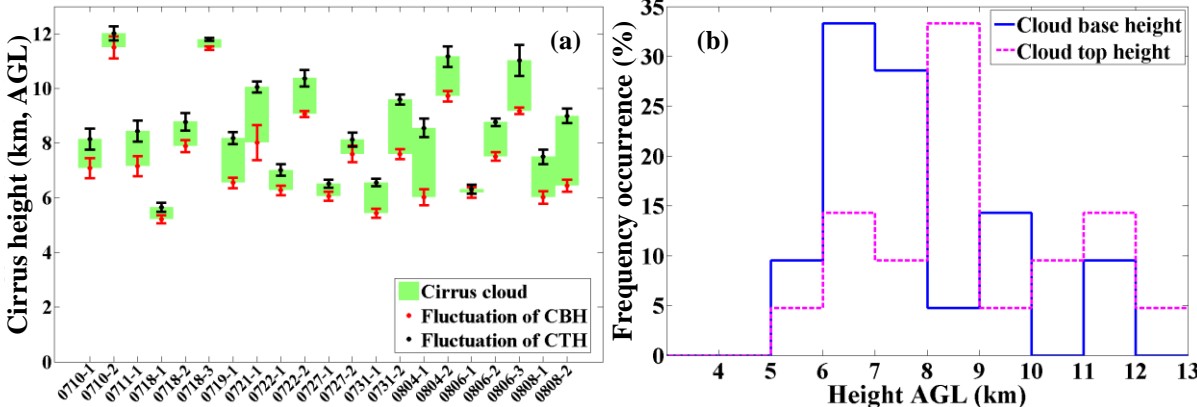

**Figure 3: (a) Cirrus structure and (b) the histogram of cloud base/top heights measured by WACAL over Naqu during the period of July and August.**

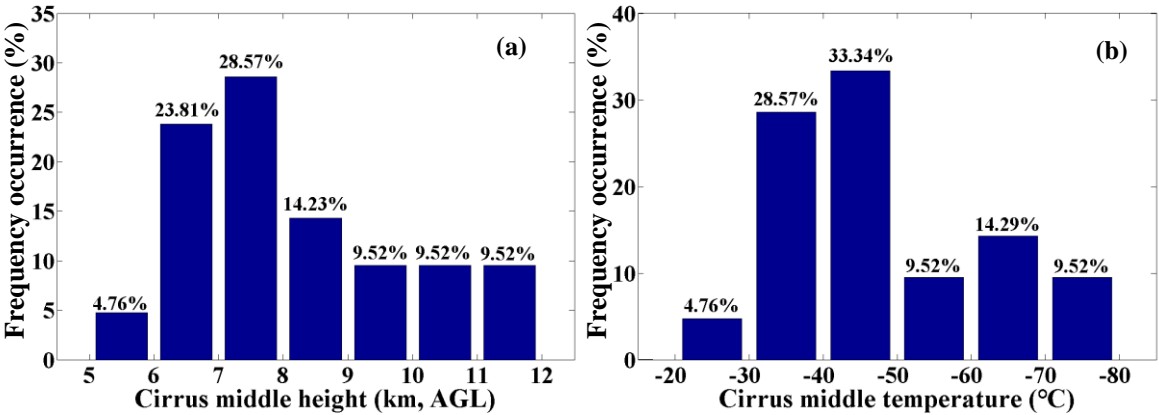

5    **Figure 4: (a) Cirrus middle height occurrence frequency and (b) temperature occurrence frequency.**

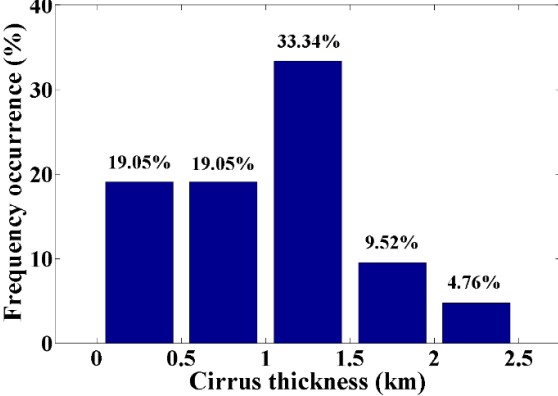

**Figure 5: Cloud thickness measured during the experiment.**





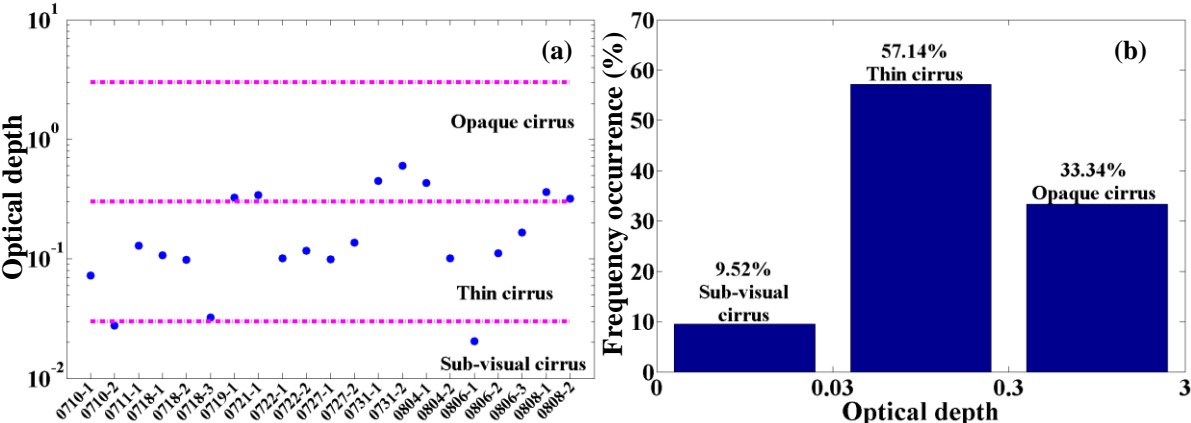

**Figure 6: (a) Distribution of optical depth and the classification, (b) Occurrence frequency of different types of cirrus clouds.**

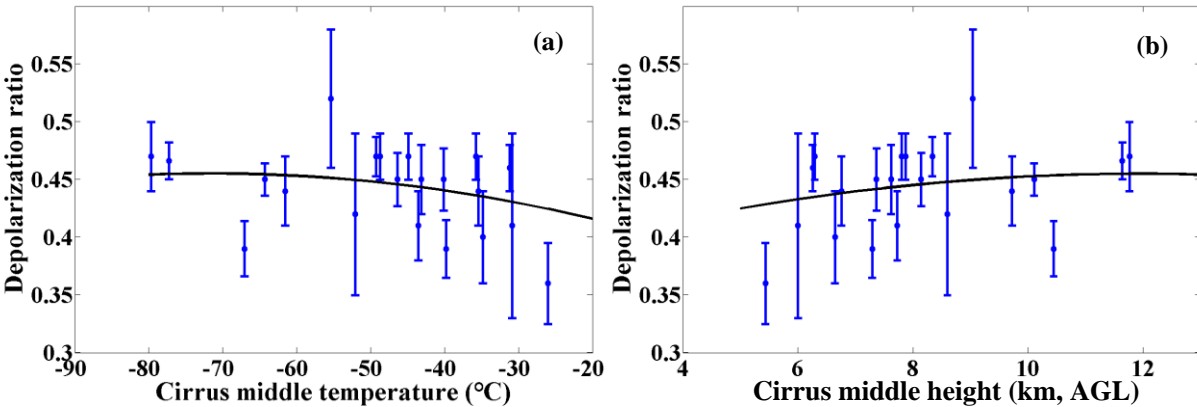

**Figure 7: (a) Scatter diagram of depolarization ratio and temperature, (b) Scatter diagram of depolarization ratio and cirrus middle**

5   **height. The black lines indicate the corresponding fitting curves.**

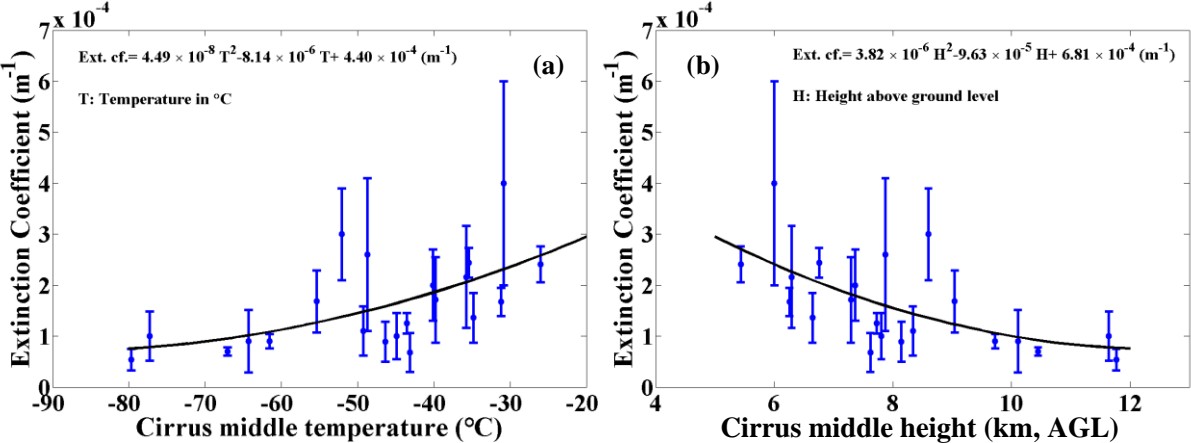





**Figure 8: (a) Scatter diagram of extinction coefficient and temperature, (b) Scatter diagram of extinction coefficient and cirrus middle height. The black lines indicate the corresponding fitting curves and the fitting formulas are also provided.**

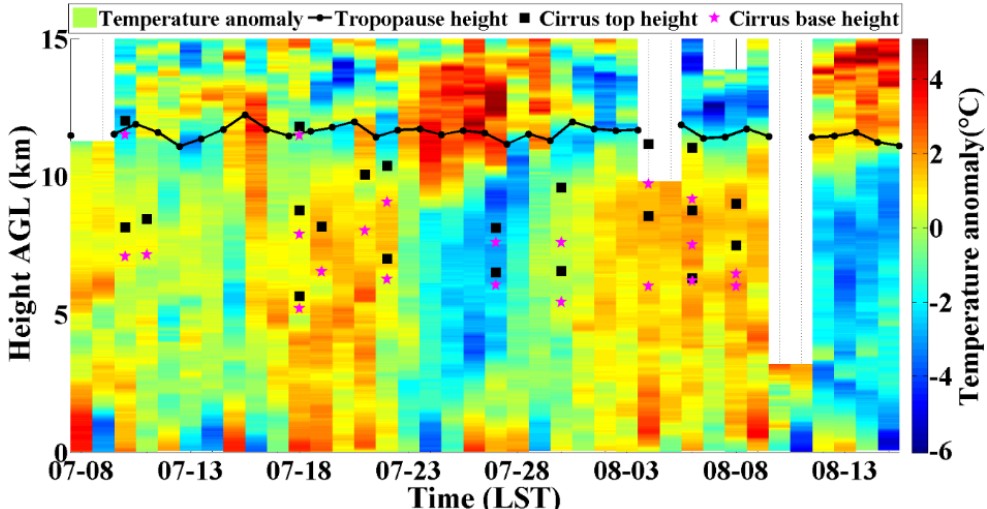

**Figure 9: Temperature anomalies during July and August 2014.**

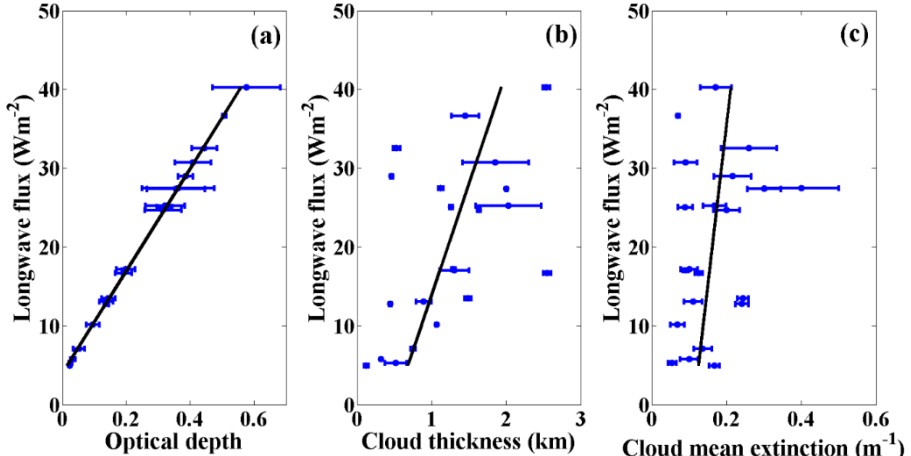

**Figure 10: Relationships between optical depth, cloud thickness, extinction coefficient and longwave flux.**





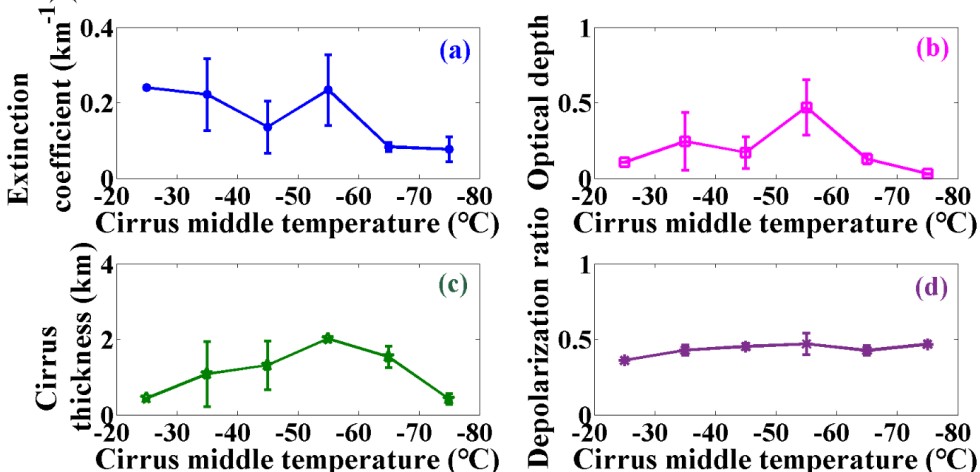

**Figure 11: (a) mean extinction coefficient (b) optical depth (c) thickness and (d) depolarization ratio on 10 °C intervals of cloud temperature.**