# Peer review of "Optical and Geometrical Properties of Cirrus Clouds over the Tibetan Plateau Measured by Lidar and Radiosonde Sounding at the Summertime in 2014"

_Atmospheric Measurement Techniques, 2017_

## Referee Comment (RC1) · Anonymous Referee #1 · 30 Jan 2018

The authors present results from the TIPEX III campaign conducted at Naqu (Tibetan Plateau, China) for two months from a sophisticated lidar system and radiosonde soundings. The purpose of the study is the investigation of cirrus properties and their radiative forcing. The relative small set of data (21 cases only) of course limits the value of the study, especially as more extended climatologies are already existing (for other sites). Consequently, for "compensation" of this shortcoming the description of the methods and the discussion of the findings must be precise and convincing, and a discussion of the uncertainty of the results is mandatory. In the present version of the

paper this is however not sufficiently provided. Moreover a clear conclusion ("what do we learn from this study?") and an outlook are missing.

As a result, major revisions are mandatory.

A few detailed comments (without typos, grammar, etc.) in the order of appearance:

1. Throughout the paper: Linear particle depolarization ratio $\longrightarrow$ particle linear depolarization ratio

2. Page 2, line 15: Move IPCC-citation two lines up.

3. 2/20: Ansmann et al. (1992) do not cover polarization lidars. Choose an additional reference.

4. 3/4: "The optical depth of clouds is a key parameter...": What about the single scattering albedo?

5. 3/9: "lidar signals correctly represented the scattering...": What does this mean? I assume that the SNR is larger than a certain threshold so that the Rayleigh signal can be measured.

6. 4/8: What is the reason for four telescopes? Give a short explanation.

7. 4/19: "$\pm$ 5%": absolute or relative?

8. 5/5: "With the help of the WACAL, the optical and geometrical properties of cirrus clouds can be obtained." Be more precise: which parameters were actually determined and what is their accuracy. Is it possible to retrieve the lidar ratio from the 355/387 nm channels? In the following the authors assume a prescribed (constant) lidar ratio.

9. 5/7: How are the ceilometer data used in this study? In contrast to WACAL it seems to me that they provide continuous measurements, so this could be a great opportunity to improve the statistics of the cirrus bottom height, top height, extent – maybe even a rough estimate of the optical depth (after calibration, from integrated backscatter if the lidar ratio is prescribed anyway) can be derived.

10. 5/17: Briefly explain how $P_{bg}$ and $P_{noise}$ are determined.

11. 5/21–23: "the maximum optical depth": what is "maximum" referring to? Explain how "with these thresholds, the dust and volcanic ash layers ... can be filtered out". Was this actually applied, i.e. was volcanic ash present at that site and that time?

12. 6/8: Explain $G$. It could make sense to repeat the equations from Freudenthaler et al. (2009), but then a discussion on the accuracy of the terms in (2) and (3) should be included. At least the overall accuracy of the derived $\delta_p$ and $\delta_v$ should be discussed.

13. 6/17: Is it necessary to repeat the equation from Ansmann et al.? Rather discuss why the relatively large lidar ratio of 28 sr was selected (in the literature typically lower values are given), and what the influence of an error in $S_{aer}$ on the retrieved optical depth and $\delta_p$ is. By the way: Subscripts "p" and "aer" for "aerosols" or "particles" should be synchronized.

14. 7/1–7: Please rephrase this paragraph: in its present form the purpose of the procedure is hard to understand and it is just copied from the Wu et al. (2015) paper.

15. 7/15: "Moreover, the particle size of cirrus clouds which.." Include 1 or 2 sentences explaining the underlying concept and how it influences the multiple scattering (MS) correction. Later in the paper the MS-issue is more or less ignored,

so what is the reason for the relatively broad discussion of MS? Moreover I am missing a (range of) value(s) of the MS-factor, its accuracy and its influence on the optical depth. It is surprising that an equation for the very well known extinction coefficient (4) is given but no equation how the MS-correction factor is applied.

16. 8/21: How is a "measurement case" defined? A period of $n$ hours with a temporal resolution of $m$ minutes? An average over the night? How can the investigation be extended with the ceilometer measurements (improvement of the "climatology")?

17. 9/5: If only one case is shown it must be explained how representative it is. Is it a "typical" case? Better show a few more examples, or report a summary of all cases.

18. 9/8: The error bars of $\delta_p$ are missing. Is it realistic to compare values above and below 7 km in view of the (large?) uncertainty (see also line 13)?

19. 9/15: The authors point out the variability of the ice crystal shape: How does it influence the lidar ratio (assumed to be constant in the paper)?

20. 9/19: "6.02 ±0.60 km": Is this the uncertainty or the temporal variability? Explain and add information on the missing. Is it reasonable to express the average with two decimal places?

21. 9/20: What is "corresponding" referring to? Is the mean extinction coefficient with or without MS-correction?

22. 10/2: Explain the base of the statistics covered in Section 3.2. Is it based on 21 cases only? How is each case treated (as a whole or with a temporal resolution of $m$ minutes)? Why is a very coarse resolution of only 1 km considered? Why are the ceilometer data neglected?

23. 10/5: How long is the period over which the "fluctuations" are determined, i.e. how long was each individual cirrus existing? Same duration for all?

24. Throughout the paper: don't use two decimal places for the percentages!

25. 10/9: Here one would expect information on the vertical extent. So think about moving the corresponding section from below to this place and/or combine Figs. 3–5 to one figure with (a) – (e).

26. 10/24: Better use $\text{km}^{-1}$

27. 11/2: Be sure that for all references the duration and the site of the corresponding study is mentioned. E.g., if Goldfarb et al. (2001) consider 384 nights it is more representative than a 21 night data set.

28. 11/23: Again, this statement implies that the lidar ratio change.

29. 11/23: "According to Fig. 7, it ...": Considering the very large error bars (or ranges of variability?) such a strong statement should be relaxed.

30. 12/5: This statement is valid for other wavelengths as well.

31. 12/10: If Platt is referenced please note that he suggested a cirrus lidar ratio of 18.2 sr.

32. 12/13: "Ext.cf." should be replaced by $\alpha$ in the equation! Please explain why such a fitting curve makes sense: As $\alpha$ is an extensive quantity it is not obvious that it depends on temperature (alone). Surely, particle shape and size are temperature dependent (and thus extinction cross section) but the extinction coefficient is expected to depend on water vapor concentration as well.

33. 12/14: "Due to the stability of the larger cirrus particle radius". What do you mean?

34. 12/19: Define "anomalies". With respect to what?

35. 13/2: "appearance stages, the temperatures were higher...": which temperatures? ("temperature below the cloud"?)? Please be more specific.

36. 13/7: "the Rossby waves can be recognized ...": Is this just a message or a finding from the measurements (or analysis of supplementary data)?

37. 13/14: "the ice particle size by the method described ..." : A 1–2 sentence synopsis would be nice, in particular including information about the inherent assumptions and the sensitivity of the results on these assumptions.

38. 14/6: "with a mean value of 0.44 $\pm$ 0.037": 0.037 seem to be the variability. What about the accuracy (see previous comments as well)?

39. 14/9: "The corresponding gradients are also calculated and are presented in Fig. 7 and Fig. 8." This statement is not very useful in the summary. Here results and conclusions should be presented. This is valid for the whole section!

    As ceilometers are mentioned in the paper as one of the measuring systems (and as they recently are becoming more widespread) one would expect a discussion of the benefit of these automated and continuous ceilometer measurements: what are the pros and cons? Consider the corresponding literature (at least some overview papers). A discussion/recommendation can be beneficial for the readership: what kind of information is obtained when ceilometers (networks?) replace the advanced (expensive) lidar.

---

## Referee Comment (RC2) · Anonymous Referee #2 · 5 Feb 2018

General remarks

The paper presents Raman lidar and radiosonde measurements of cirrus clouds over the Tibetan Plateau during the time from July to August 2014. The instrumentation, calibration and data processing methods are outlined and the derived cloud properties are discussed. Specifically, cloud geometric properties, depolarization ratio, extinction ratio, and temperature are presented. Furthermore the relationship between these cloud properties and radiative properties as well as temperature is examined. In this work, already documented and established methods and instrumentation is used, to

gather information on relevant cirrus properties in a region that, according to the authors, has not been investigated with ground-based lidar before. Field campaigns like this help to broaden our knowledge on cirrus around the globe and its results are well worth publishing. However, AMT is "dedicated to the publication and discussion of advances in remote sensing, as well as in situ and laboratory measurement techniques". As this paper does not involve new instrumentation or methods of data analysis, the submission at another journal is recommended.

Before publication, major revisions are needed. To make this work beneficial for the community, the authors should focus more on what the distinct features of cirrus clouds over the Tibetan Plateau at the specific time of measurement are, compared to other regions and seasons. When in chapter 3.3 the dependence of depolarization ratio on cloud middle temperature is presented, only the behavior of an ill-motivated fitting curve is discussed. But in parts, the data shows huge variability that seems to contradict the general remarks on the fitting curve (see specific remarks also). These up to now unmentioned deviations should be examined and discussed in more detail. Also the suggested interplay of Rossby waves and cloud appearance in chapter 3.4 seems to be very promising. But this section consists mostly of the presentation of measurements and a bold statement about the convective origin of the measured clouds, without references and sufficient back up from the data. Here a more thoroughly investigation and discussion is needed. The authors should avoid the excessive enumeration of measured and derived values in the abstract, in chapter 3, and in the summary (a table might help). This distracts the reader and after all, these numbers might not be generally valid due to the comparably short measurement period. Rather, more effort on drawing clearer conclusions on the distinguishing features should be made.

Following these remarks, I do not recommend the publication in AMT. After a revision, I would like to highly encourage the authors to submit their manuscript at a less technique-oriented and more science-oriented journal.

Specific remarks

1.: p.5, l.8: "cirrus occurred almost at nighttime and before dawn" What do you mean here? "Almost allways at nighttime and before dawn" or shortly before nighttime (almost nighttime)? Maybe give also the local time to make things clearer.

2.: p.5, l.9: "more conductive " than what? Why is occurrence at night conductive for your measurement? How many radio sondes were launched? When were they launched? Regularly or only when cirrus was there? How many hours between soundings? Maybe put that information in chap. 1.2 already.

3.: p.6, l.19: "S is assumed as a constant and independent of the range", How is it justified and what consequences does that have for the derived ext. coeff.?

4.: p.8, l.16: How is the opt. depth derived? Which assumptions do you make on particle size? How do the assumptions influence the retrieved variable?

5.: p.8, l.23: Radiosondes are launched twice a day. There are up to 6 hours between launch and lidar measurements. How does this affect the analysis of data?

6.: p.9, l.6: "there are two cirrus clouds layers were observed" -> "two cirrus cloud layers were observed"

7.: p.9, l.8: "For instant" -> "For instance"? Why do you choose this case?

8.: p.10, l.4: "Fig 3(a) shows the cirrus cloud structure..." it would be better to write "shows the horizontal extent of cirrus"

9.: p.10, l.5: "makes the details of cirrus clouds, much clearer." Which details do you mean? It is actually less detailed than Fig 2. Also, do not say "it makes thinks much clearer." Why even include Fig 3a)? It is not discussed in the text anymore and is redundant with Fig. 9. Just leave it out.

10.: p.10, l.5: "fluctuation", what is this? Please clarify.

11.: p.10, l.15: I see a skewed distribution but not a skewed normal distribution.

12.: p.11, l.1-7: "Several investigations on the statistics of cirrus clouds have been accomplished..." First present your results than put them in perspective to earlier findings.

13.: p.11, l.16: "statistical uncertainty", what do you mean? Please clarify.

14.: p.11, l.16: "fitting curves" Why choose this type of curve?

15.: p.11, l.21: "We propose that with the decrease of temperature... of the cirrus clouds" This might be correct generally (on a climatological view) but this idea is already well established. More interesting would be to know why at -68°C you only have 0.37 and at -55°C you have up to 0.55. How could these "anomalies" depend on specific influences at this site, season and weather?

16.: p.11, l.24: "... lower altitude is much more sensitive to the ambient temperature." Your data base is rather small, can you really back up this statement confidently with your data, especially given the fact that there are huge fluctuations in data colder than -50°C?

17.: p.11, l.24: "Under the condition of low temperature, since the existence of the stability of the larger cirrus particle radius,..." What do you mean here (A particle radius cannot be stable or unstable)? Do you want to say that at cold temperatures, particles are bigger and therefore depolarization ratio is not changing with temperature anymore? How do you support that statement (references)? Why should particles be bigger at lower temp, where less water vapor is accessible?

18.: p.12, l.3: "stat. uncertainty" What is that exactly? Would it be better to show e.g. 10th and 90th percentiles of data at that temperature?

19.: p.12, l.15: "... stability of the larger cirrus particle radius..." again what do you mean? Please clarify and back up with references.

20.: p.12, l.19: "temperature anomalies", how are they calculated? Temperature deviation from what?

21.: p.12, l.22: "less than 1,5 km above tropopause" Is this due to the time between cirrus measurement and temperature measurement? Or due to measurement uncertainties? Please comment on that. Also, might it be possible for cirrus to occur above the tropopause?

22.: p.13, l.2: add the height information: "the temperatures from 5 to 11 km, were higher..."

23.: p.13, l.3: You indicate that cirrus formation may be related to deep convective activity. Would lower liquid clouds be a common feature in that case? How would that influence your record of cirrus occurrence? I suspect that blocking of the lidar by the liquid clouds could be a problem. Please comment on that.

24.: p.13, l.8-11: "Considering the inherent characteristics of strong local heating ... to the formation of cirrus clouds." This is a very interesting point but up to now it is only a statement. I think it might be worthwhile investigating this in more detail. Provide references, check the connection with Rosby waves from other sources such as satellite data. What kinds of clouds are measured here? Are they convective outflow cirrus from farther away? Do they form directly in the convection above the lidar? How does the radiative exchange between cold anomaly at the tropopause and lower layers work to start deep convection?

25.: p.13, l.12-22: This section about radiative properties could be put into context of the above stated "deep-convection-theory". Do the cirrus clouds influence the start and stop of deep convection? The particle size is modelled, how do the used assumptions influence the result?

26.: p.14, l.15-21: Consider putting this section (with Fig. 11) into the main body of this work, before the summary.

27.: Figure 9: Please use a diverging colormap with only three color hues, e.g. "red-white-blue" (http://colorbrewer2.org/#type=diverging&scheme=RdBu&n=11)